# Characterization of Two Novel *Bacillus thuringiensis* Cry8 Toxins Reveal Differential Specificity of Protoxins or Activated Toxins against Chrysomeloidea Coleopteran Superfamily

**DOI:** 10.3390/toxins12100642

**Published:** 2020-10-05

**Authors:** Changlong Shu, Guixin Yan, Shizhi Huang, Yongxin Geng, Mario Soberón, Alejandra Bravo, Lili Geng, Jie Zhang

**Affiliations:** 1State Key Laboratory for Biology of Plant Diseases and Insect Pests, Institute of Plant Protection, Chinese Academy of Agricultural Sciences, Beijing 100193, China; clshu@ippcaas.cn (C.S.); yanguixin@caas.cn (G.Y.); simjue@126.com (S.H.); yongxin_Geng@hotmail.com (Y.G.); llgeng@ippcaas.cn (L.G.); 2Instituto de Biotecnología, Universidad Nacional Autónoma de México, Cuernavaca 62250, Mexico; mario@ibt.unam.mx (M.S.); bravo@ibt.unam.mx (A.B.)

**Keywords:** *Bacillus thuringiensis*, *cry8* genes, Coleopteran, insecticidal activity, Cry protoxin, dual mode of action

## Abstract

Scarabaeoidea and Chrysomeloidea insects are agriculture-destructive coleopteran pests. Few effective *Bacillus thuringiensis* (Bt) insecticidal proteins against these species have been described. Bt isolate BtSU4 was found to be active against coleopteran insects. Genome sequencing revealed two new *cry8* genes in BtSU4, designated as *cry8Ha1* and *cry8Ia1*. Both genes expressed a 135 kDa protoxin forming irregular shape crystals. Bioassays performed with Cry8Ha1 protoxin showed that it was toxic to both larvae and adult stages of *Holotrichia parallela*, also to *Holotrichia oblita* adults and to *Anoplophora glabripennis* larvae, but was not toxic to larval stages of *H. oblita* or *Colaphellus bowringi*. The Cry8Ia1 protoxin only showed toxicity against *H. parallela* larvae. After activation with chymotrypsin, the Cry8Ha1 activated toxin lost its insecticidal activity against *H. oblita* adults and reduced its activity on *H. parallela* adults, but gained toxicity against *C. bowringi* larvae, a Chrysomeloidea insect pest that feeds on crucifer crops. The chymotrypsin activated Cry8Ia1 toxin did not show toxicity to any one of these insects. These data show that Cry8Ha1 and Cry8Ia1 protoxin and activated toxin proteins have differential toxicity to diverse coleopteran species, and that protoxin is a more robust protein for the control of coleopteran insects.

## 1. Introduction

The Coleoptera (beetles) insect order contains more described species than any other animal group, with over 380,000 named species [1,2], comprising almost 40% of described insects and nearly 30% of all animal species. Among these beetles, the species from Scarabaeoidea and Chrysomeloidae superfamilies are considered as the most destructive coleopteran pests, which cause severe damages in agriculture, horticulture, and forestry. The insecticide formulations based on *Bacillus thuringiensis* (Bt) insecticidal proteins, as well as transgenic plants expressing Bt insecticidal proteins are applied as sustainable control methods for some species of these coleopteran pests [3,4]. However, comparing to the beetle’s diversity, the currently discovered variability of Bt insecticidal proteins effective against coleopteran insects is scarce. Currently, only few members in Cry1, Cry3, Cry6 (now APP6), Cry7 Cry8, Cry22 (now XPP22), Cry23/Cry37 (now MPP23/XPP37), Cry34/Cry35 (now GPP34/TPP35), Cry55 (now XPP55), Cyt1, Cyt2, Sip (now MPP5), and Vip1/Vip2 subgroups of Bt protein families were reported to be toxic against certain coleopteran insects [5,6]. Hence, screening of Bt strains and identification of novel insecticidal proteins active against coleopteran insects, especially proteins that display a broad insecticidal spectrum against Chrysomeloidea and Scarabaeoidea is still needed for the effective control of these pests.

Bt insecticidal crystal proteins (ICPs) are synthetized as protoxins which, once ingested by the insect, are required to be solubilized, and proteolytically activated in the insect midgut [7,8]. The activated toxin binds to membrane receptors and lyses midgut epithelial cells by forming pores that cause cell swelling. Some of the Cry toxins with potential activity are only toxic after in vitro solubilization and activation with commercial proteases. For example, the toxicity of Cry7Aa towards coleopteran insect larvae was revealed only after an in vitro solubilization and activation with proteases [9]. For this reason, it is recommended to perform bioassays with both protoxins and activated toxins to detect insecticidal activity of a new Bt ICPs, especially against coleopteran larvae. In this report, we screened a new Bt strain, BtSU4, with insecticidal activity against *Holotrichia parallela* and we cloned two novel *cry8*-type genes from this strain. Furthermore, the expression, solubilization, activation and toxicity assays of the two novel ICPs indicated that the new Cry8Ha1 protein has a broad insecticidal spectrum important for the control of several important coleopteran pests.

## 2. Results

### 2.1. Cloning and Sequence Analysis of cry8Ha1 and cry8Ia1 from BtSU4

After BtSU4 genome sequencing and insecticidal gene annotation, two genes coding for proteins with similarity to Cry8 family were discovered. The two Cry8 proteins consisted of 1199 and 1194 amino acids, respectively. The two novel proteins were subsequently named as Cry8Ha1 and Cry8Ia1 (https://www.bpprc.org). 

After CDD search, the core region of the two novel insecticidal proteins were identified. The domains I, II, and III of Cry8Ha1 protein were located from amino acids 94 to 290, 300 to 510 and 512 to 657, respectively. While for Cry8Ia1 protein these domain regions were located from amino acids 106 to 288, 296 to 516, and 517 to 665, respectively. BLAST analysis results showed that some domains of the two Cry8H proteins have high similarity to other proteins from the Cry family such as Cry3, Cry1B, or Cry1I, rather than to Cry8 members. 

Phylogenetic analysis were performed with the complete protoxin or the toxin core sequences of these two Cry8 proteins and compared with different Cry proteins (Figure 1). These analyses showed that protoxin sequences of Cry8Ha1 and Cry8Ia1 clustered in a single branch with the other Cry8 protoxins as expected. In contrast the sequence of the toxin cores of these proteins were found in different branches. The toxin core of Cry8Ha1 toxin was found clustered in the same branch with Cry3 proteins together with Cry8Ba, Cry8Pa, Cry8Ca, and Cry8Ja, while Cry8Ia1 was found grouped with Cry8G, Cry8K, and Cry7G toxins (Figure 1). Suggesting that these two novel proteins Cry8Ha1 and Cry8Ia1 may have differential specificity. 

We also performed phylogenetic analyses of the different domains. Analysis of domain I sequences from several Cry proteins showed that this region is distributed in three main clusters (Appendix A). Cluster A includes 11 different Cry8 proteins clustered with domain I sequences from Cry1B and Cry1I (Appendix A, Cluster A); the second group (Appendix A, Cluster B), includes domain I sequences from Cry8Ha1 and Cry8Ia1 that were found to be grouped with other nine Cry8 and Cry3 proteins; the third group (Appendix A, Cluster C) includes six members, forming a cluster including more distant Cry8 proteins (Appendix A). The sequences from domains II and III showed larger variation, where the Cry8Ha1 and Cry8Ia1 were located in different clusters. For domain II sequences six different clusters were identified, the Cry8Ha1 was located in Cluster A (Appendix A, Cluster A), which included sequences of domain II from Cry1B and Cry1I proteins, while Cry8Ia1 is found in cluster E with other Cry8 members such as Cry8Ib1, Cry8Ga1, Cry8Ka1, and Cry8Kb1 (Appendix A, Cluster E). Finally, for domain III sequences, they were classified in seven clusters, where the domain III of Cry8Ha1 is found in Cluster A, which includes sequences from Cry1I, Cry1B and Cry3 proteins (Appendix A, Cluster A), while Cry8Ia1 is found in Cluster E together with other Cry8 proteins, such as Cry8Ib1, Cry8Sa1, Cry8Ta,1 and Cry8La1 (Appendix A, Cluster E). These analyses show that Cry8Ha1 toxin is more closely related with Cry3, Cry1B, and Cry1I.

### 2.2. Differential Toxicity of Cry8Ha1 and Cry8Ia1 Protoxins and Activated Toxins Against Different Coleopteran Insects

To express the *cry8Ha1* and *cry8Ia1* genes, primers were designed according to their gene sequences, amplified from the total DNA of BtSU4 strain and cloned into pSTK shuttle vector as described in Materials and Methods. The acrystalliferous HD-73^-^ strain was transformed with these constructions by electroporation, and the resulted recombinant strains HD8H and HD8I were used to express these proteins and determine their toxicity against different coleopteran insects as described below. The morphology of the crystals produced by BtSU4, HD8H, and HD81 strains was analyzed in spore/crystal suspensions observed under scanning electron microscopy (SEM). Figure 2 shows that BtSU4 had typical Bt spores and relatively small irregular crystals (indicated by arrow). The ICPs accumulated in BtSU4 crystals were composed of 135 kDa protein, as revealed by SDS-PAGE electrophoresis (Figure 2E).

The SEM images of Figure 2 showed that both Cry8Ha1 and Cry8Ia1 crystals inclusions accumulated forming large irregular crystals (Figure 2B,C, indicated by arrows). SDS-PAGE analysis showed that the expressed Cry8Ha1 and Cry8Ia1 protoxins were resolved also as 135 kDa bands (Figure 2E), similar to the ICPs molecular weight band observed for BtSU4 strain.

To compare the toxicity of protoxins with that of activated toxins of both Cry8Ha1 and Cry8Ia1 proteins, the protoxins of Cry8Ha1 and Cry8Ia1 were activated by chymotrypsin digestion. We decided to activate Cry8 toxins with chymotrypsin since it was previously shown that this protease was highly efficient for activation of Cry3Aa toxin that is toxic to coleopteran larvae in contrast to trypsin protease [10]. After digestion with chymotrypsin, the activated toxins were again analyzed by SDS–PAGE electrophoresis, showing that the Cry8Ha1 is activated into a 70 kDa protein fragment after 2 h of treatment with the protease. The 70 kDa protein fragment was relative stable since it remained after 24 h of treatment, although some of the 70 kDa fragment was further degraded into a smaller fragment with a molecular weight about 55 kDa (Figure 3). In contrast, the protoxin of Cry8Ia1 after protease treatment showed two main fragments of 97 and 66 kDa (Figure 3).

The toxicity of the two Cry8Ha1 and Cry8Ia1 protoxins was first analyzed against different insects (Table 1, Figure 4). The data showed that the Cry8Ha1 protoxin was toxic to both *H. parallela* larvae and adults. This protoxin was also toxic against *H. oblita* adults, but not to *H. oblita* larvae neither to *C. bowringi* larvae (Table 1). Analysis against *A. glabripennis* larvae showed that Cry8Ha1 protoxin was also toxic to the larvae of this insect (Figure 4). The Cry8Ha1 treated larvae showed a severe effect in their growth, they remain small, lack mobility and looks severely damaged, after few days of treatment they died. The image of Figure 4B shows these effects. In contrast, the Cry8Ia1 protoxin, showed toxicity only against *H. parallela* larvae and did not show toxicity to any the other insect tested in this study (Table 1). We then analyzed toxicity of the activated toxin samples, chymotrypsin activation caused the complete loss of toxicity of Cry8Ha1 to *H. oblita* adults and seems to reduce the activity against *H. parallela* adults when compared to the Cry8Ha1 protoxin, although these data are not significant since their fiducial limits overlapped. The activation of Cry8Ha1 toxin with chymotrypsin resulted in enhanced toxicity against *C. bowringi* larvae (Table 1). The activated Cry8Ia1 toxin showed no toxicity to any insects tested in this study.

## 3. Discussion

The Cry8 toxin family has been reported to have toxicity against coleopteran insects and until now 60 different Cry8 proteins (including 2 Cry8-like proteins) that belong to 28 holotypes by the new Bacterial Pesticidal Protein Resource Center (BPPRC) (https://www.bpprc.org) [6]. Cry8 proteins have been shown to be mainly toxic to Scarabs, but some patents claimed that certain Cry8 proteins also show toxicity to Chrysomeloidea insects [11]. The larvae and adults of both Scarabs and Chrysomeloidea insects feed on plants, thus it is important to find effective strategies to control both stages of these insects. Until recently, as shown in Figure 5, the Cry8Ab, Cry8Ca, Cry8Da, Cry8Db, Cry8Ea, Cry8Ga, Cry8Na, and a Cry8-like protein have been confirmed toxic to six main harmful scarabs. Among these proteins, the Cry8Da, Cry8Db and Cry8Ga proteins have shown to be effective against both adults and larvae stages of the insects [12,13]. In this study, we show that the protoxin from the novel Cry8Ha1 showed toxicity against both adults and larvae of *H. parallela*. In contrast, this Cry8Ha1 protoxin was only toxic to the adults of *H. oblita*, but not to the larvae. In addition, Cry8Ha1 was toxic to the larvae of *A. glabripennis*. The protoxin of the new Cry8Ia1 was only toxic to the larval stages of *H. parallela.* The reason for the differential toxicities of these protoxins observed in adults or larvae stages of certain species is not known, but one alternative could be related to a differential expression of Cry8 midgut receptors in both developmental stages. This hypothesis remains to be analyzed. 

The phylogenetic analysis of Cry8Ha toxin core sequence showed that this proteins is more related to Cry3 proteins and the analysis of the different domains of Cry8Ha1 and Cry8Ia1 proteins confirmed that Cry8Ha1 is more related to the coleopteran active proteins Cry3, Cry1B, and Cry1I that have been shown to be specifically toxic to Chrysomeloidea insects [5]. Our data suggest that the novel Cry8Ha1 and Cry8Ia1 proteins have evolved from different origins, since these Cry8Ha1 and Cry8Ia1 proteins do not cluster together. Only Domain I of both Cry8Ha1 and Cry8Ia1 proteins clustered together with domain I from Cry3 proteins. Domain II of Cry8Ha1 is more related to domain II from Cry1B and Cry1I proteins; while Domain III of this protein is closer to domain III from the Cry1B, Cry1I, and Cry3 coleopteran active proteins. In contrast Domains II and III of Cry8Ia1 are not related to the corresponding domains of other coleopteran active toxins, only to other Cry8 proteins, such as Cry8Ib and Cry8Ga. Based in these close phylogenetic relationships with other Cry toxins active against Chrysomeloidea insects we decided to determine the toxicity of the two novel Cry8Ha1 and Cry8Ia1 proteins against some Chrysomeloidea pests, such as *C. bowringi* and *A. glabripennis*. For *C. bowringi*, Cry8Ha1 was effective, but the toxicity was revealed only with the Cry8Ha1 activated toxin after in vitro activation. Previously, a similar observation was made with the Cry7Aa1 protein against Colorado potato beetle where only the in vitro activated Cry7Aa1 protein showed to be toxic [9]. It has been shown that the content of midgut proteases may have great variations between different insects orders, for example, the main digestive proteases of Lepidoptera and Diptera are serine proteases, whereas those of Coleoptera are mainly cysteine and aspartic proteases [14]. Therefore, it is possible that the Cry8Ha1 is not processed correctly by the *C. bowringi* midgut proteases but the in vitro processing could activate the toxin properly revealing its toxicity against *C. bowringi* larvae. In the case of *A. glabripennis*, the Cry8Ha1 protoxin was effective against *A. glabripennis* larvae, which is a Cerambycidae insect that harms many forests plants. Until recently, only Cry3Aa protein was shown to has toxicity to some Cerambycidae species, including *Phytoecia rufiventris* Gautier, *Apriona germari*, as well as *A. glabripennis* [15,16,17]. The discovering of new insecticidal proteins with different amino acid sequences that may have different modes of action is important to have additional tools for the control of Cerambycidae and Chrysomeloidea insects. The toxicity of Cry8Ha1 to two different Cerambycidae and Chrysomeloidea species suggests that other Cry8 proteins could also potentially show toxicity to different coleopteran species.

In addition, we also find that the in vitro solubilization and activation was adverse for Cry8Ha1 activity to scarab adults and also for its toxicity against the larval stages of *H. parallela,* since a reduction in toxicity was observed with the activated toxin. These data could indicate that the treatment with chymotrypsin inactivates the Cry8Ha1 protoxin or that two different mechanisms exist for protoxin or activated toxin, as previously suggested [15]. The fact that Cry8Ha1 activated toxins increases is toxic effect against *C. bowringi* larvae indicates that chymotrypsin treatment did not inactivate Cry8Ha1. Rather, the data indicate that both Cry8Ha1 protoxin and activated toxin may have independent toxicity pathways in the different insect species analyzed. In the case of Cry1Ab and Cry1Ac toxins, it has been shown that protoxins and activated toxins exert toxicity by independent pathways, where two different oligomer prepores with different pore formation activity are formed depending if protoxin or activated toxin bind to cadherin receptor [18]. Furthermore, it has been shown that the C-terminal region of Cry1Ab protoxin provides additional binding sites for alkaline phosphatase (ALP) and aminopeptidase N (APN) receptors providing a higher binding affinity of the protoxin to the gut membrane which contributes to the higher toxicity of Cry1Ab protoxin compared to the activated toxin [19]. It remains to identify the different protoxin or activated Cry8Ha1 binding proteins in the different insects analyzed and in the different developmental stages. However, an alternative explanation could be that the differential toxicity observed between Cry8Ha1 protoxin or activated toxin is related to the differential proteases that are present in the different insects analyzed. In any case, our data point out that the differential toxicities of protoxin or activated toxin should be taken into account to decide which form of the toxin should be expressed in transgenic plants, depending on the target pest. 

## 4. Conclusions

In this paper, we characterized two new insecticidal proteins, Cry8Ha1 and Cry8Ia1. In addition to the larvicidal activity against *H. parallela*, the Cry8Ha1 showed to be toxic to the adults of *H. parallela* and *H. oblita*, and to the larvae of some Cerambycidae insects such as *C. bowringi* and *A. glabripennis*. Considering the sequence cluster characterization and the insecticidal spectrum of Cry8Ha1, this study not only provides new tools to control coleopteran insects, but also a valuable material for the ongoing investigation of insecticidal specific evolution, as well as potential strategies for further toxin improvements.

## 5. Materials and Methods

### 5.1. Bacteria Growth Conditions and Plasmids

The BtSU4 strain used in this study was isolated from soil samples in Hebei province, China. HD-73^-^ a crystal negative Bt strain, was used as recipient strain for the expression of the novel Cry8 proteins. Bt strains were incubated for 3 days in cross-baffled flasks at 30 °C with shaking at 230 rpm in PB medium (0.5% peptone, 0.3% beef extract; pH 7.2) for expressing the *cry8* genes. *Escherichia coli* DH5α strain was used for common transformation, while *E. coli* SCS110 strain was used to produce nonmethylated plasmid DNA for Bt transformation. The *E. coli* strains were grown at 37 °C in Luria-Bertani medium (LB: 1% tryptone, 0.5% yeast extract, 1% NaCl; pH 7.0). Plasmids derivative from pSTK [27] were constructed for cloning and expression of the genes. Kanamycin (50 μg mL^−1^) was added to the media, when appropriate, for selection of *E. coli* and Bt antibiotic resistant strains. 

### 5.2. Gene Cloning and Sequence Analysis 

In order to clone ICPs genes from BtSU4 strain, the Roche/454 technology was used to sequence the complete genome of this strain. After genome assembling, performed by Roche 454’s Newbler, the contig00204 (GenBank accession number MT905434) was detected, containing two ICPs genes by using the Basic Local Alignment Search Tool (BLAST, https://ftp.ncbi.nlm.nih.gov/blast/executables/blast+) and an in-house Bt toxin genes database. The two *cry*-type genes were deposited in GenBank with the following accession numbers AY897354 and EU381044. The new Cry proteins were annotated by using the Conserved Domains Database (CDD) search (https://www.ncbi.nlm.nih.gov/cdd/). The Phylogenetic analyses of the amino acid sequences were conducted by Neighbour-joining [28] and MEGA X [29]. 

### 5.3. Protein Expression, Extraction and Activation

Primers were designed to clone the novel ICPs genes. SU4_5 (5′-GGA ATT CGA TGA GTC CGA ATA ATC AGA AT-3′) and 8H_3 (5′-CGC GTC GAC TTA CAT TTC TTC TAC AAT CAA TTC-3′) primers were designed to amplify *cry8Ha1* gene, while SU4_5 and 8I_3 (5′-CTC ATT TCT TCT ACA ATC AAT TCT ACA CTG TC-3′) primers were used to amplify *cry8Ia1* gene. A KOD DNA polymerase and PTC-100 Peltier Thermal Cycler (MJ Research) was used for PCR, comprising 30 cycles (each cycle was composed of 1 min at 94 °C, 1 min at 54 °C, and 4 min at 68 °C) followed by 10 min incubation at 68 °C. The amplified full-length *cry8Ha1* and *cry8Ia1* genes were inserted into the *Eco*RI-*Sal*I sites and *Eco*RI-*EcoI*CRI sites of the pSTK shuttle vector, respectively. The plasmids purified from *E. coli* DH5α strain were then transformed into *E. coli* SCS110 for producing nonmethylated recombinant plasmids. Then, the expression vectors containing *cry8Ha1* and *cry8Ia1* genes were introduced into the crystal negative Bt strain HD-73^-^ by electroporation [27]. The positive transformant colonies were incubated in LB medium supplemented with kanamycin at 30 °C until sporulation. The spore/crystal mixtures were collected and washed twice in 1.0 M NaCl and sterile distilled water successively. The final suspension was analyzed by sodium dodecyl sulfate-polyacrylamide gel electrophoresis (SDS-PAGE) 10% acrylamide gel.

The crystal/spore mixtures were subsequently solubilized in a pH 10.2 buffer containing 50 mM Na_2_CO_3_, 50 mM EDTA, 3% 2-mercaptoethanol, incubated on a rotary shaker at 150 rpm at 0 °C for 4 h. The supernatant soluble protoxins were separated by centrifugation at 13,500× *g* for 10 min and the protoxins were precipitated by adding 4 M Sodium acetate-Acetic acid (NaAc-HAc) buffer (pH 4.5) until the pH reached 5.0 and then incubated at 4 °C for 4 h. The protoxin pellet was collected by centrifugation at 13,500× *g* and washed with pre-chilled distilled water three times to remove NaAc and HAc, and then dissolved in 50 mM Na_2_CO_3_ (pH 10.2). The protoxins were activated by treatment with chymotrypsin (Sigma) (10:1 w/w). The digestion was carried out in 50 mM Na_2_CO_3_ buffer (pH 10.2) at 37 °C for 2, 12 or 24 h. Reactions were stopped by chilling in ice. The digested proteins were then boiled 3 min with protein-loading dye and analyzed on a 10% acrylamide SDS–PAGE. 

### 5.4. Scanning Electron Microscopy (SEM) Examination 

After sporulation in LB medium, Bt spores and crystals were harvested by centrifugation at 13,500× *g* for 1 min. The pellets were suspended and washed twice in distilled water. SEM micrographs were taken according to the method described before [27]. The suspension was smear on glass slide and placed on aluminum stubs, dried, and fixed in 1% OsO4. Then, the spore crystal mixtures were sputter-coated with gold in an IB-5 ion coater (EIKO Engineering) for 4 min. The SEM micrographs were taken on a Hitachi S4800 digital scanning microscope (Hitachi High-Tech, Japan) at a voltage of 12 kV

### 5.5. Insects, Rearing Conditions and Bioassay

Adults of *H. oblita* and *H. parallela* were collected from Baoding and Cangzhou, Hebei province, China. These adults were reared in plastic boxes filled with 5 cm thick sieved soil and fed with elm (*Ulmus pumila* L.) leaves, at 25 ± 0.5 °C, 16L: 8D photoperiod condition and 60% humidity. The eggs were collected and hatched in the same conditions. 

For bioassay activity against adults, we used the leaf-dipping method. Before feeding, the cleaned elm leaves were soaked in Cry protoxin protein solutions, and then the surface liquid was dried at room temperature. In each treatment, ten adults were feed with the dipped leaves, and then, we replaced the protein dipped elm leaves and check insect mortality every day. These assays were done in triplicate for each protein concentration, using a two-fold dilution protein concentration series (0.03125 mg/mL, 0.0625 mg/mL, 0.125 mg/mL, 0.25 mg/mL, 0.5 mg/mL, 1.0 mg/mL, and 2.0 mg/mL). For bioassays against larvae, we used a procedure modified from Shu (2009) [20]. Five-day larvae were feed with a mixture of potato with spores/crystal suspension. Firstly, the potato was shred, washed, air dried and then mixed with Bt spore/crystal water suspension; then, the mixture was further mixed with sieved soil and divide evenly into four six-well culture plates. In each treatment, 24 larvae were feed separately in each well performed in triplicate and mortality was determined after 14 day.

The adults of *A. glabripennis* were collected from Beijing, China, and reared in plastic boxes filled with pieces of willow branches (*Salix babylonica*), at 25 ± 0.5 °C, 16L: 8D photoperiod condition, 60% humidity. The adults feed on the bark and lay their eggs in the bark. The eggs hatch and larvae bioassay were performed on artificial diet. To prepare the artificial diet, we made willow bark powder and willow xylem powder. Branches with a diameter of 5–6 cm from the growing willow were used to separate the willow bark and xylem. Then, we washed, dried, ground, and sieved (80 meshes) the bark and xylem, respectively. The resulted powders were ready for use. The artificial diet was produced by mixing 60 g willow bark powder, 30 g willow xylem powder, 22.5 g agar, 10 g yeast powder, 1.5 g sorbic acid, 1.5 g methyl 4-hydroxybenzoate, 1.5 g ascorbic acid, and 520 mL sterile water, and sterilized at 121 °C for 20 min. To perform bioassays, the Cry proteins were mixed with the artificial diet and divided into a 24-well culture plates, with final protein concentrations of 100 μg/g, and 300 μg/g; then, 24 two-day larvae were feed separately in each well in triplicate in a dark condition, and mortality was determined after 10 days.

The *C. bowringi* Baly population was reared in the lab using rape leaves. The bioassays were performed in triplicate against larvae by using the leaf dipping method. Before feeding, the cleaned rape leaves were soaked in Cry protein solutions with two-fold dilution series of protein concentrations of 0.125 μg/mL, 0.0625 μg/mL, 0.03125 μg/mL, 0.01563 μg/mL, and 0.007815μg/mL, and then the surface liquid was dried at room temperature. In each treatment, 20 larvae were feed with the dipped leaves, at 25 ± 0.5 °C, 16L: 8D photoperiod, 60% humidity, the mortality was determined after two days.

As mentioned above three replicates were performed for each treatment for all bioassays. The 50% lethal concentrations (LC_50_) were determined by Probit analysis. For all leaf dipping method, 0.005% Triton X-100 was added to increase fluid ductility and allowing uniform protein dip.

## Figures and Tables

**Figure 1 toxins-12-00642-f001:**
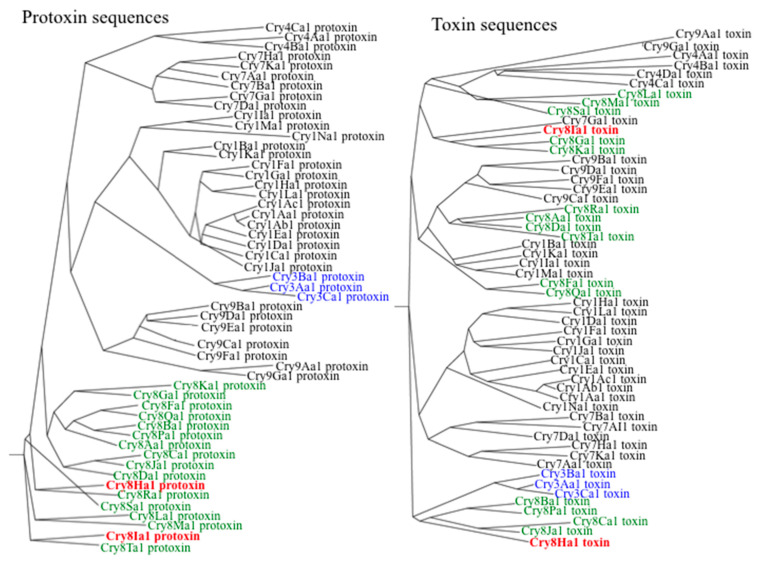
Phylogenetic analysis of the protoxin or toxin core amino acid sequences from different Cry proteins. These analyses involved 52 amino acid sequences. Evolutionary analyses were conducted in Clustal Omega and Neighbour-joining Phylogenetic analysis (Madeira et al. 2019). The Cry8Ha1 and Cry8Ia1 proteins were labeled in red letters.

**Figure 2 toxins-12-00642-f002:**
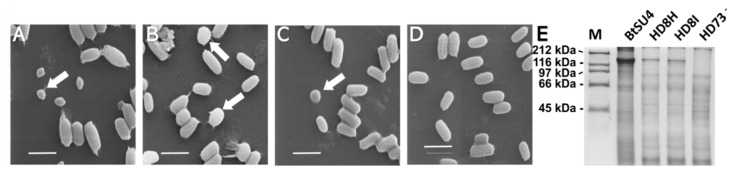
SEM observation and Electrophoretic analysis of Bt spore and crystal mixtures. Panels (**A**–**D**), SEM images of the spore and crystal mixtures of Bt strain BtSU4, HD8H, HD8I and crystal negative strain HD-73^-^. The scale bar in panels A, B, C and D was 2 μm. Panel (**E**), Sodium dodecyl sulfate-polyacrylamide gel electrophoresis (SDS-PAGE) profiles of these spore and crystal mixtures. M, Bio-Rad high-molecular-mass protein marker. The other lanes were total protein samples from Bt strain BtSU4, HD8H, HD8I and crystal negative strain HD-73^-^, respectively.

**Figure 3 toxins-12-00642-f003:**
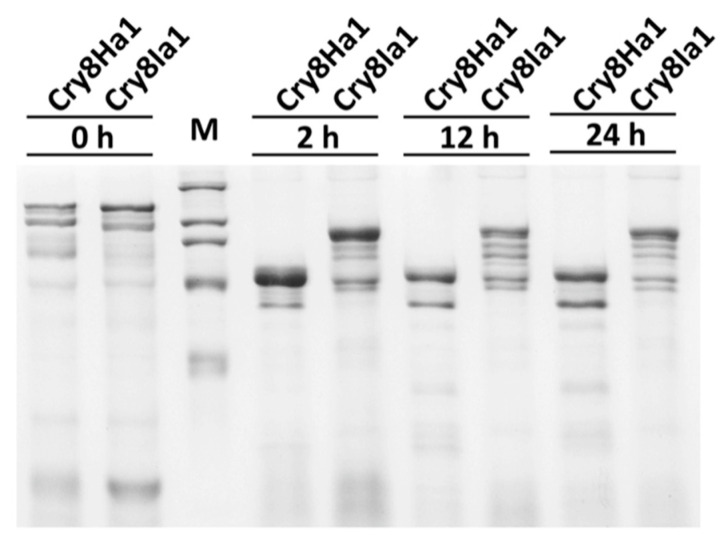
SDS-PAGE protein profile of Cry8Ha1 and Cry8Ia1 proteins, after digestion with chymotrypsin protease. “M” means the marker, and the molecular weight of the bands are 212, 116, 97, 66 and 45 kDa, respectively.

**Figure 4 toxins-12-00642-f004:**
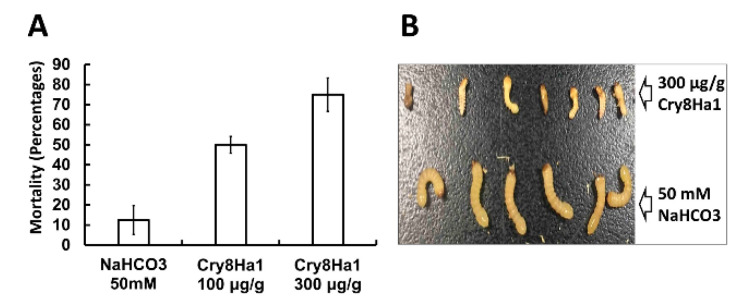
The bioassay of Cry8Ha1 protoxin against *Anoplophora*
*glabripennis*. (**A**): the mortality of *A. glabripennis* treated by different dose of Cry8Ha1 protoxin. This experimented was performed in triplicate. For each repeat, 24 larvae were tested; (**B**): Morphology of toxin-treated *A. glabripennis* larvae.

**Figure 5 toxins-12-00642-f005:**
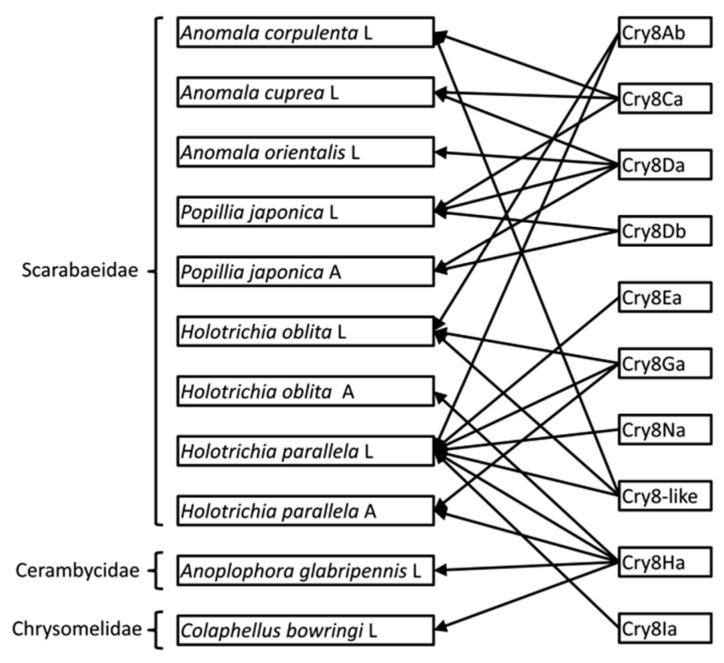
Insecticidal activity data of Cry8 proteins published before and in this work [12,20,21,22,23,24,25,26]. The “A” and “L” labels after the insect name means the “Adult” or “Larvae” stages of those insects.

**Table 1 toxins-12-00642-t001:** The LC_50_ Cry8Ha1 and Cry8Ia1 against *Holotrichia oblita*, *Holotrichia parallela*, and *Colaphellus bowringi*.

Sample	*H. oblita*		*H. parallela*		*C. bowringi*
	Adult LC_50_ Values in mg/mL (Confidence Limits)	Larvae 10^8^ CFU/g	Adult LC_50_ Values in mg/mL (Confidence Limits)	Larvae LC_50_ Values in 10^8^ CFU/g (Confidence Limits)	Larvae LC_50_ Values in mg/mL (Confidence Limits)
Cry8Ha1-Pro	0.99 (0.62–2.05)	NA	0.16 (0.00–0.57)	218.90 (68.21–661.10)	NA
Cry8Ia1-Pro	NA	NA	NA	85.03 (30.59–176.08)	NA
Cry8Ha1-Act	NA	NT	0.66 (0.20–12.36)	NT	0.02 (0.01–0.02)
Cry8Ia1-Act	NA	NT	NA	NT	NA

Note: the NA = no activity; NT = not tested; “-Pro” = protoxin; “-Act” = activated toxin; “CFU” = the colony-forming unit of crystal-spore mixture.

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
