# Peer review of "Characterization of Two Novel Bacillus thuringiensis Cry8 Toxins Reveal Differential Specificity of Protoxins or Activated Toxins against Chrysomeloidea Coleopteran Superfamily"

_toxins, 2020, doi:10.3390/toxins12100642_

Round 1

Reviewer 1 Report

The paper contains data that may be interesting to the readers, who focus on bacterial toxins. It is mostly descriptive. It is clearly written. In my opinion, the materials and methods must be developed. First of all, the experimental design needs to be well described. Also, the SEM techniques should be described: the type of the microscope should be named, the procedure of fixation, etc. Also, SEM pictures in Figure 2, presenting SEM pictures, need the scale bars. 

Figure 4 mentions the morphological changes. They should be described in the results.

Author Response

The paper contains data that may be interesting to the readers, who focus on bacterial toxins. It is mostly descriptive. It is clearly written. In my opinion, the materials and methods must be developed. First of all, the experimental design needs to be well described. Also, the SEM techniques should be described: the type of the microscope should be named, the procedure of fixation, etc. Also, SEM pictures in Figure 2, presenting SEM pictures, need the scale bars. 

Response: We improved the materials and methods, and described SEM techniques as requested by the reviewer. The following sentences were added to the text “For electron microscopy, the suspension was smear on glass slide and placed on aluminum stubs, dried, and fixed in 1% OsO4. Then, the spore crystal mixtures were sputter-coated with gold in an IB-5 ion coater (EIKO Engineering) for 4 min. The SEM micrographs were taken on a Hitachi S4800 digital scanning microscope (Hitachi High-Tech, Japan) at a voltage of 10 kV.” We also add “The scale bar in panels A, B, C and D was 2 μm.” In the figure legend of Fig 2.

Figure 4 mentions the morphological changes. They should be described in the results.

Response: the morphological changes of the larvae after intoxication with Cry8Ga1 are now described in the text as requested. We add the following sentence in the text “The Cry8Ha1 treated larvae showed a severe effect in their growth they remain small, lack mobility and looks severely damaged, after few days of treatment they died. The image of figure 4 B shows these effects.”

Reviewer 2 Report

In this paper, the authors created recombinants of Cry8Ha and Cry8Ia and examined their insecticidal spectra. I think it is a meaningful study and worth publishing. However, you should consider revising the following points.

Line 57

"PB( )" is not required.

Line 70

I could not search "EF465532" in the NCBI database. I recommend AY897354 because I think it is the same as EF465532.

Line 93, NaAc-HAc buffer

Please use the abbreviation after defining it.

Line 131

What is the reason for writing in red?

Line 150

The website is already closed.

Figure 3

Add the molecular weight on the band of the marker.

Table 1

Align the number of significant digits.

Figure 4A

Error bars are required for bar charts. It seems that you have only experimented once, so it is a good idea to make a table of the number of animals used, the number of responding animals, and the mortality rate.

Line 246

It is better to delete the site notation of “Bacillus thuringiensis Delta-Endotoxin Nomenclature Committee” because it will be skipped to the BPPRC site.

That's all

Author Response

In this paper, the authors created recombinants of Cry8Ha and Cry8Ia and examined their insecticidal spectra. I think it is a meaningful study and worth publishing. However, you should consider revising the following points.

Line 57

"PB( )" is not required.

Response: We deleted it.

Line 70

I could not search "EF465532" in the NCBI database. I recommend AY897354 because I think it is the same as EF465532.

Response: This accession number was revised.

Line 93, NaAc-HAc buffer

Please use the abbreviation after defining it.

Response: This was revised.

Line 131

What is the reason for writing in red?

Response: This was a mistake and we changed it.

Line 150

The website is already closed.

Response: We changed it to https://www.bpprc.org/

Figure 3

Add the molecular weight on the band of the marker.

Response: We add the following sentence in the figure legend of figure 3 ” “M” means the marker, and the molecular weight of the bands are 212, 116, 97, 66 and 45 kDa, respectively.”

Table 1 Align the number of significant digits.

Response: Revised.

Figure 4A

Error bars are required for bar charts. It seems that you have only experimented once, so it is a good idea to make a table of the number of animals used, the number of responding animals, and the mortality rate.

Response: We add error bars for the bar charts. We performed the experimented once with three repeats. For each repeat, 24 larvae were tested. This information was included in the figure legend of Figure 4.

Line 246

It is better to delete the site notation of “Bacillus thuringiensis Delta-Endotoxin Nomenclature Committee” because it will be skipped to the BPPRC site.

Response: We deleted it as suggested.